# Managing Pediatric Asthma Exacerbations: The Role of Timely Systemic Corticosteroid Administration in Emergency Care Settings—A Multicentric Retrospective Study

**DOI:** 10.3390/children11020164

**Published:** 2024-01-26

**Authors:** Luna Antonino, Eva Goossens, Josefien van Olmen, An Bael, Johan Hellinckx, Isabelle Van Ussel, An Wouters, Tijl Jonckheer, Tine Martens, Sascha Van Nuijs, Carolin Van Rossem, Yentl Driesen, Nathalie Jouret, Eva Ter Haar, Sabine Rozenberg, Els Vanderschaeghe, Susanne van Steijn, Stijn Verhulst, Kim Van Hoorenbeeck

**Affiliations:** 1Laboratory of Experimental Medicine and Pediatrics, Department of Pediatrics, Faculty of Medicine and Health Sciences, University of Antwerp, 2610 Antwerp, Belgium; luna.antonino@uantwerpen.be (L.A.); stijn.verhulst@uantwerpen.be (S.V.); 2Centre for Research and Innovation in Care, Department of Nursing Science and Midwifery, Faculty of Medicine and Health Sciences, University of Antwerp, 2610 Antwerp, Belgium; eva.goossens@uantwerpen.be; 3Department of Public Health and Primary Care, KU Leuven, 3000 Leuven, Belgium; 4Department of Patient Care, Antwerp University Hospital, 2610 Antwerp, Belgium; 5Department of Family Medicine and Population Health, University of Antwerp, 2610 Antwerp, Belgium; josefien.vanolmen@uantwerpen.be; 6Department of Pediatric Pulmonology, Hospital Network Antwerp, 2020 Antwerp, Belgium; an.bael@zna.be (A.B.); carolin.vanrossem@zna.be (C.V.R.); yentl.driesen@zna.be (Y.D.); eva.terhaar@zna.be (E.T.H.); sabine.rozenberg@zna.be (S.R.); els.vanderschaeghe@zna.be (E.V.); susanne.vansteijn@zna.be (S.v.S.); 7Translational Science, Department of Immunology and Inflammation, 2610 Antwerp, Belgium; 8Department of Pediatrics, General Hospital Klina (AZ Klina), 2930 Brasschaat, Belgium; johan.hellinckx@klina.be; 9Department of Pediatrics, General Hospital Voorkempen (AZ Voorkempen), 2390 Malle, Belgium; isabelle.vanussel@uza.be (I.V.U.); an.wouters@emmaus.be (A.W.); 10Department of Pediatric Pulmonology, Antwerp University Hospital, 2610 Antwerp, Belgium; nathalie.jouret@uza.be; 11Department of Pediatrics, GasthuisZusters Antwerp, 2610 Antwerp, Belgiumtine.martens@zas.be (T.M.); sascha.vannuijs@zas.be (S.V.N.)

**Keywords:** asthma, asthma exacerbation, children, emergency service, hospital, corticosteroids

## Abstract

Background: Asthma is the most prevalent chronic respiratory condition in children. An asthma exacerbation (AE) is a frequent reason for emergency department (ED) visits. An important step in the management of a moderate to severe AE is the administration of systemic corticosteroids (SCS) within 1 h after ED presentation. This study aimed to determine the timing of SCS administration and correlate this with the length of stay and oxygen therapy duration and to explore factors predicting timely administration. Methods: This study used a retrospective multicenter observational design based on electronic medical records review. Children aged < 18 years, presenting to the ED with a moderate to severe AE were included. Results: 205 patients were included. Only 28 patients received SCS within 60 min after ED arrival. The median time to SCS administration was 169 min (Q_1_ 92–Q_3_ 380). A correlation was found between timing and oxygen treatment duration (r = 0.363, *p* < 0.001) and length of stay (r = 0.368, *p* < 0.001). No patient characteristics predicted timely SCS administration. Conclusions: Three in four children who presented with a moderate to severe AE at the ED did not receive SCS within the first hour. A prolonged timing of SCS administration correlated with a prolonged length of stay and extended need for oxygen support.

## 1. Introduction

Pediatric asthma is the most prevalent diagnosed chronic respiratory disease in children [1,2,3]. It is an inflammatory disorder of the airways associated with bronchial hyper-responsiveness, reversible airflow limitation and symptoms like dyspnea, coughing, wheezing and tightness of the chest [1,4,5,6]. In 2019, asthma affected about 262 million people globally. In Belgium, about 517,200 children were diagnosed in 2021 [7,8], and 7% of children aged between 6 and 20 years, are currently diagnosed with asthma [8,9]. Asthma has different phenotypes, which can make it difficult to differentiate from other airway problems, making it very challenging to determine the ‘true’ prevalence of asthma for this age group [8,10]. In children above the age of 6 years, the prevalence of asthma appears to be higher in boys (8.1% vs. 5.9% in girls), while during adolescence, it shifts to girls (7.5% vs. 6.2% in boys) [8,10].

An asthma exacerbation is one of the most common reasons for a visit to the emergency department (ED) in children [7,8,9,10]. Such an exacerbation usually occurs as a response to a trigger or exposure to an external agent (e.g., viral respiratory infection, allergen exposure, air pollution or seasonal changes) and/or patient non-adherence to prescribed controller medication [2,9]. According to the current set of international guidelines, the severity of an asthma exacerbation is determined by the symptoms experienced by the patient [9]. The treatment of such an exacerbation is described by several international protocols, including the GINA guidelines [9,11,12,13]. One of the most important steps in the management of moderate to severe asthma exacerbations is the timely administration of systemic corticosteroids (SCS) within the first hour after the initial presentation of the patient [9,14,15,16]. In a prior retrospective study conducted across more than 4000 patient ED visits in the USA, it was found that the average duration between patients’ arrival and the administration of SCS was 93 min [17]. Additionally, this study revealed that only half of the sample received systemic corticosteroids within 60 min [3,17]. A study performed by the same research group demonstrated that administering corticosteroids to pediatric patients with asthma presenting at the ED within 1 h of triage was associated with a 25 min mean decrease in ED length of stay (LOS), as well as a 4.8% decrease in admission rates and ED return rates for patients presenting with a moderate to severe asthma exacerbation [3,17]. To date, empirical studies investigating the implementation of the current set of guidelines on timely SCS administration, its consequences, and potential predictors are scarce, and an in-depth understanding is mandatory to improve the quality of care.

This study aims to improve the care for children admitted to the ED due to an acute asthma exacerbation. A retrospective multi-center study was performed aiming (i) to measure the timing of systemic corticosteroid administration to children presenting at the ED with a moderate-to-severe asthma exacerbation, (ii) to compare the timing of administration to the guideline-based standard of administration within the first hour of ED presentation, (iii) to correlate this timing with several outcome parameters including length of stay and duration of respiratory support, and (iv) to explore patient-related characteristics predicting timely SCS administration.

## 2. Materials and Methods

### 2.1. Design and Setting

A retrospective multicentric observational study using medical chart review was conducted at the EDs of the Antwerp Pediatric Asthma Network (APAN), in Belgium. APAN consists of several departments of pediatrics in the Antwerp region (Belgium), consisting of one tertiary and six secondary care facilities, aiming to improve care for children with asthma in general through joined research and quality improvement projects. The following hospitals are formal partners of this network and are participating: University Hospital Antwerp, ZNA Middelheim, ZNA Jan Palfijn, ZNA Paola Children’s Hospital, GZA St-Augustinus, GZA St-Vincentius and AZ Klina.

In the ED department, a child is seen by a specialist nurse and triaged (red: acute, orange: very urgent, yellow: urgent, green: standard, blue: not urgent). After triage, the necessary tests are carried out and the child will receive the first emergency treatment. The examinations are carried out by an emergency doctor, who may call in a specialist pediatrician. Upon admission, the patients’ medical history is consulted in the EMR if the patient was already known in the hospital.

### 2.2. Study Population

The study population included children under the age of 18 who presented with a moderate-to-severe asthma exacerbation in an ED visit at one of the participating centers. Patients had to have received systemic corticosteroids and be admitted to the hospital. A moderate asthma exacerbation is defined as the worsening of asthma symptoms, an FEV_1_ > 50% best or predicted, and no features of acute severe asthma [9]. A severe asthma exacerbation is defined as any one of: FEV_1_ ≤ 50% best or predicted, respiratory rate ≥ 30/min, accessory muscles being used, heart rate ≥ 120/min, O_2_ saturation < 90%, and inability to complete sentences in one breath [9]. An ED visit is defined as a visit where the primary ED diagnosis was a first asthma exacerbation or a recurrent asthma exacerbation. The GINA guidelines recommend the administration of systematic corticosteroids (SCS) in all patients presenting with an asthma exacerbation, except patients presenting with a mild exacerbation who experience prompt relief through administration of a bronchodilator [9]. Thus, patients were excluded if they presented with a mild asthma exacerbation and if they received systemic corticosteroids within 24 h prior to their respective ED visit because of their potential influence on symptoms, symptom relief and the choice to administer SCS by the treating physicians. Patients who had a diagnosis of bronchiolitis or had a fever at presentation were also excluded. A consecutive sample was composed of all eligible patients who presented at the ED between 1 January 2019 and 31 December 2020.

### 2.3. Data Collection

Data were collected through chart review from patients’ electronic medical records (EMR) at the ED and pediatrics departments during the study period. A set of socio-demographic and clinical patient characteristics, as well as timing of medication administration and length of hospital stay (LOS) were collected from the different EMR (see Appendix A).

All patients admitted to the ED with an asthma exacerbation were retrospectively identified over the course of the study period through an electronic search of the EMR. For each visit, using a standardized data extraction form, discrete fields in the EMR were used to obtain the time of arrival (i.e., timing of ED patient registration), patient characteristics, and descriptors of the ED visit (i.e., mode of arrival (self vs. ambulance) and triage data (presenting vital signs, triage acuity)). Information about the duration of oxygen administration/respiratory support and hospital admission (i.e., pediatric department vs. PICU (pediatric intensive care unit) stay) was also extracted from the EMR. The respective timings of ordering and administering systemic corticosteroids were obtained from the electronic prescription programs. The length of hospital stay was calculated by extracting the date of ED admission and the discharge date. Time to systemic corticosteroid administration was calculated from the date and time of ED arrival to the date and time of the first SCS administration. A delayed administration was defined as an interval > 60 min [9].

### 2.4. Ethical Considerations

This study was performed in accordance with the Declaration of Helsinki [18]. The Institutional Review Board of the Antwerp University Hospital/University of Antwerp (coordinating center) and the respective local institutional review boards of the participating centers approved study execution. Patient data were pseudonymized according to GDPR guidelines [19].

### 2.5. Statistical Analysis

Data analysis was performed using IBM SPSS Statistics, version 28.0 (IBM Corp., Armonk, NY, USA). Normality and homogeneity of the data were checked using the Kolmogorov–Smirnoff test. Data were considered significant at *p* < 0.05. Descriptive statistics were performed for all included cases and are presented as means ± standard deviations (SD) if data were normally distributed and medians with interquartile range (IQR) if data were not normally distributed. Nominal variables were described using absolute numbers (n) and percentages (%). Two variables had cases with differing values: LOS (4 cases) and duration of oxygen (1 case). Analyses were performed with and without these cases to determine if they were outliers. Since no significant differences in test results were observed when analyses were performed with and without those respective cases, the final set of analyses were performed on the entire sample.

Children were, based on their age at ED admission, categorized as being either aged ≤ 71 months or ≥72 months; as per international guidelines, an official diagnosis of asthma can only be made when children reached the age of 6 years (i.e., ≥72 months).

The primary outcome was the prevalence of delayed steroid administration (i.e., time of administration > 60 min). Four categories of delayed SCS administration were differentiated: administration between 60 and 90 min after admission, administration between 91 and 120 min of admission, administration between 121 and 150 min after admission and administration after ≥151 min. Potential differences between children in terms of their sex, age, reason of admission and home medication across these four respective categories were tested using a χ^2^-test, Fisher’s Exact test or Kruskal–Wallis test.

The median time to SCS administration was compared using a Mann–Whitney U test in relation to the mode of transportation to the hospital (i.e., own transportation vs. via ambulance), need for oxygen supportive therapy (i.e., yes or no) and the department of admission (i.e., pediatric department or PICU).

Furthermore, using a Mann–Whitney U test, the median age (in months), LOS (in days) and duration of oxygen (in days) were compared in relation to whether or not SCS administration was delayed. The sex and need for oxygen (i.e., yes or no) were compared between children with either timely or delayed SCS administration using a χ^2^-test or Fisher’s Exact test.

The time to SCS administration was correlated, using the Spearman correlation coefficient, with the duration of oxygen support and the overall length of stay. A multivariable logistic regression analysis was used to explore which patient-related factors independently predicted timely SCS administration. Results were reported using odds ratios (OR) and 95% confidence intervals (95% CI).

## 3. Results

### 3.1. Sample Characteristics

The EMR of all patients admitted to the ED due to an asthma exacerbation were screened between 1 January 2019 and 31 December 2020. A total of 205 patients were found to be eligible, and the sample characteristics are described in Table 1.

The sample comprised 117 boys (57.1%) and 88 girls (42.9%). Patients had a median age of 70 months (Q1–Q3: 38.5–112.5 months), and 106 children (51.7%) were younger than 6 years. The number of patients admitted to the ED varied over the years. In 2019, 116 children (56.6%) were admitted, this number decreased to 89 children (43.4%) in 2020. The median length of stay within the hospital was 3 days, with a range from 2 to 4 days.

The hospitals were coded to ensure anonymity. In Table 2, the characteristics per center are described, such as the number of patients per center, the median time to SCS administration, the median length of hospital stay and the median duration of oxygen.

### 3.2. Timely Administration of SCS

In this study, 28 patients (13.7%) received systemic corticosteroids within 60 min after ED arrival. The median time to administration of SCS was 169 min (Q1 92 min–Q3 380 min). A delayed SCS administration occurred in 177 patients (86.3%). Within the group of children experiencing delayed SCS administration, 12.4% had an administration time between 61 and 90 min, 9.6% between 91 and 120 min, 13% between 121 and 150 min and 65% after more than 151 min. There were no significant differences in the distribution of children over these categories of delayed SCS administration in terms of their sex, age, reason for admission and if the patient had taken medication at home before ED admission (see Table 3).

### 3.3. Correlates of SCS Administration

A significant difference was observed in the median time of SCS administration between patients who were admitted to a pediatric ward and those admitted to PICU (m_e_ 174 min vs. m_e_ 109 min, *p* = 0.039). A significant difference was observed in median LOS between patients whose SCS administration was timely and those whose was delayed (m_e_ 2 days vs. m_e_ 4 days, *p* < 0.001). For the median duration of oxygen support (n = 117), a significant difference was found between patients who received timely SCS and those who received delayed SCS (m_e_ 1 day (IQR 1) vs. m_e_ 3 days (IQR 9), *p* < 0.001).

No significant differences were found in terms of median age (m_e_ 65.5 months (IQR 166) vs. m_e_ 71 months (IQR 195), *p* = 0.816) and sex (χ^2^ = 0.175, *p* = 0.675) between patients with either delayed or timely SCS administration. The median duration of SCS administration did not differ significantly between patients who did or did not receive oxygen support therapy upon ED arrival (m_e_ 158 min vs. m_e_ 178 min, *p* = 0.224), arrived using their own transportation or by ambulance (m_e_ 168.5 min vs. m_e_ 173 min, *p* = 0.977), did or did not need oxygen therapy (χ^2^ =0.000, *p* = 0.994) or depending on the department of admission (*p* = 0.186). The time of SCS administration was moderately correlated with the duration of oxygen support (in days, n = 117, rho = 0.363, *p* < 0.001) and with the length of hospital stay (rho = 0.368, *p* < 0.001).

### 3.4. Predictors of Timely and Delayed SCS Administration

A multivariable logistic regression was used to analyze the relationships among sex, age, mode of patient transportation to the ED, medication at home, oxygen therapy and either timely or delayed SCS administration. None of these variables were found to significantly predict the odds of a timely or delayed SCS administration (See Table 4 and Table 5).

## 4. Discussion

Asthma stands as the most prevalent chronic respiratory condition in children globally, often leading to visits to emergency departments (ED) due to acute exacerbations [1,2,3,7,8,9]. Timely administration of systemic corticosteroids (SCS) within the first hour of ED admission is crucial for managing moderate to severe asthma exacerbations [9,11,12,13]. To date, however, insight into the compliance of healthcare providers with this recommendation remains scarce. Furthermore, there is a need for studies investigating the consequences and predictors of (non-)timely SCS administration to further improve the quality of pediatric emergency care provided to these children. Hence, the objective of this multicenter retrospective study was to determine the adherence rate to the guideline-recommended one-hour administration time of SCS, to examine the clinical consequences of a delayed SCS administration and to explore if there was a patient profile that had a higher chance of receiving SCS in a timely manner.

The findings of this study demonstrated that more than three out of four patients had a delayed SCS administration with a median time interval between ED admission and SCS administration of 169 min (Q1 92 min–Q3 380 min). Hence, in general, most children in our study received SCS after more than two and a half hours. In a previous study conducted by Sneller et al. [3], the mean time to SCS administration was found to be 93 min, indicating that one in two patients actually did have a timely administration of SCS [3]. Hence, the adherence rate to the respective guideline-based SCS administration time was far below the results reported previously, indicating important room for quality improvement in practice.

Multiple factors can be associated with this lack of timely systemic corticosteroid administration in the ED. When considering systemic corticosteroid administration in children younger than 6 years old, a potential explanation is the uncertainty surrounding the effectiveness of SCS in this particular age group [14,20]. Since a firm diagnosis of asthma can, according to international guidelines, only be made in children starting from 6 years onwards, the treatment of asthma-like exacerbations in young children can be challenging [9,21]. The challenges experienced when trying to determine a diagnosis in children presenting at the ED with respiratory distress may contribute to a hesitant administration of SCS, which can be defined as inadequate management of asthma in this age group. Lintzenich et al. [22] conducted a study demonstrating that children between the ages of 1 and 6 years who were hospitalized for asthma-like symptoms were less likely to receive baseline treatment with inhaled corticosteroids as well as asthma-oriented patient education as compared to older children [22]. These findings highlight the potential disparities in treatment and education provided to younger children with asthma, suggesting a need for increased attention and appropriate management strategies addressing the unique challenges of diagnosing and managing asthma in this young age group (i.e., <6 years).

Secondly, the delayed administration of systemic corticosteroids could likely be attributed to the absence of a prior history of asthma, indicating that physicians may prefer to observe young children and potentially conduct additional ancillary tests before confirming a diagnosis of asthma [4,23]. This cautious approach suggests that physicians aim to gather more comprehensive information and ensure an accurate diagnosis before initiating specific asthma treatments, such as the administration of systemic corticosteroids. By closely monitoring and assessing young children, healthcare providers strive to make informed decisions regarding steroid administration, considering all relevant factors and diagnostic evidence, to ensure optimal patient care and management [4,23].

Thirdly, a significant impact on the delay in administering SCS can be attributed to physicians, who may have chosen to observe the patient’s response to the recommended bronchodilator treatments before initiating SCS therapy [3,23]. By waiting to assess the patient’s response to bronchodilators, the administration of steroids might subsequently become delayed [3,23]. This approach reflects the cautious decision making process aimed at evaluating the effectiveness of bronchodilator treatments before proceeding with additional interventions such as SCS administration [3,17].

The study of Sneller et al. [3] furthermore reported that a timely administration of SCS led to a 25 min decrease in LOS at the ED, a decrease in the return rate to the ED and a reduction in admission rates to the ED [3]. In our study, these variables were, however, neither collected nor available due to the retrospective character of data collection. However, a significant difference was observed in median LOS between patients whose SCS administration was timely or delayed (m_e_ 2 days vs. m_e_ 4 days, *p* < 0.001), as well as a significant difference in duration of oxygen therapy between patients who received either timely or delayed SCS (m_e_ 1 day (IQR 1) vs. m_e_ 3 days (IQR 9), *p* < 0.001). The time of SCS administration was moderately correlated with the duration of oxygen support (in days, n = 117, rho = 0.363, *p* < 0.001) and with the length of hospital stay (rho = 0.368, *p* < 0.001).

Traditionally, oral corticosteroids are believed to reach their maximum effect 4 to 6 h after administration, leaving some healthcare providers to prioritize other aspects of care over early corticosteroid administration [3,14,17]. However, this study challenges that notion by suggesting that administration of systemic corticosteroids could offer clinical benefits much earlier than anticipated, as demonstrated by the reduced need for oxygen support and hospital LOS. These results highlight the importance of considering early administration of SCS as a priority for optimizing patient outcomes in the context of altering their LOS and oxygen dependency. The later patients receive their SCS, the longer they tend to become hospitalized and the longer they are in need of oxygen-supportive therapy.

However, it is important to acknowledge that LOS is a complex outcome measurement affected by various factors beyond the timing of corticosteroid administration. Future studies should consider assessing additional variables as well as provider-related factors to gain a more comprehensive understanding of the factors influencing LOS. Such insights could help to develop strategies for optimizing patient care and resource utilization within the ED setting.

This retrospective study did not assess other factors that could potentially influence the prevalence of timely administration of SCS. One such factor is the administration of supplementary asthma medications that are not part of the standard treatment algorithm but are left to the discretion of the treating physician. The use of these medications may impact the timing of SCS administration. Additionally, the operational aspects of the ED, such as the number of treating physicians, the total patient load, and the presence of critically ill patients, could also influence the timing of SCS administration as well as the length of stay. Unfortunately, these data were not available based on the retrospective chart review performed as part of our study. Furthermore, variations in the timing of SCS administration could be influenced by differences in healthcare providers’ approaches, as they might have used different assessment criteria, and they might have personal preferences towards the administration of specific corticosteroids or a tendency to discharge children earlier. These individual provider factors can further contribute to variations in timely SCS administration and the LOS, but they were not explicitly assessed in this study.

Finally, our study had a limited number of cases, which makes the statistical power limited. The period in which patients were included was at the start of the COVID-19 pandemic. This can explain the reduced admission rates of children observed, especially in the year 2020. Similarly, the local air quality and air pollution in the Antwerp region may affect the prevalence of asthma in the region where the study was conducted.

### Future Research

The administration of systemic corticosteroids plays a vital role in effectively managing acute asthma exacerbations, especially in young patients. These medications are considered essential in providing effective treatment for such acute episodes. Although several guidelines recommend early SCS administration in young people presenting with an asthma exacerbation, the number of studies demonstrating the impact of timely administration in terms of clinical outcomes is, however, limited. Moreover, the implementation of clinical care pathways has proven to be beneficial in enhancing asthma management, specifically in children experiencing acute asthma exacerbations [17]. By following established clinical pathways, healthcare providers can assure standardized and optimal care provision to children during these episodes, leading to improved outcomes and better overall management of acute asthma exacerbations. Future research should include assessments of the effect of implementing such clinical care pathways for children with a moderate-to-severe asthma exacerbation in the emergency department. Furthermore, additional research into the level of knowledge and clinical insight of nurses and physicians of pharmacological guidelines regarding asthma in children is required.

## 5. Conclusions

Within an often challenging and crowded ED environment, the timely administration of systemic corticosteroids to children who present with moderate and severe asthma has shown significant benefits. Timely SCS administration leads to a shortened length of hospital stays and shorter durations of active supportive treatments, indicating its potential for optimal effectiveness. However, despite the existence of an international guideline that specifically recommends administering systemic corticosteroids within 60 min (with a standard deviation of 15 min) from triage, only one in four children met the optimal timing criteria. This highlights the need for improved adherence to the recommended timing of corticosteroid administration to achieve the benefits and outcomes for children with asthma in the ED setting. This study identified significant room for improvement in terms of the clinical and timely management of children presenting at the ED with moderate-to-severe asthma exacerbations. Future quality improvement projects focusing on achieving timely SCS administration through the training of healthcare providers and/or the implementation of a clinical care pathway are deemed mandatory.

## Figures and Tables

**Table 1 children-11-00164-t001:** Sample characteristics (n = 205).

Variables	n (%)
Sex	
Male	117 (57.1)
Female	88 (42.9)
Age in months; median (Q1–Q3)	70 (38.5–112.5)
Age category	
≤71 months (i.e., <6 years)	106 (51.7)
≥72 months (i.e., ≥6 years)	99 (48.3)
Reason of admission	
Dyspnea	12 (5.9)
First asthma exacerbation	75 (36.6)
Recurrent asthma exacerbation	118 (57.6)
Department of admission	
Children ward	188 (91.7)
PICU	17 (8.3)
Year of admission	
2019	116 (56.6)
2020	89 (43.4)
Length of stay (in days; median (Q1–Q3))	3 (2–4)
Medication at home	
Yes	75 (36.6)
No	130 (63.4)
Transport to emergency department	
Own transportation	196 (95.6)
Ambulance	9 (4.4)
Need for oxygen	
Yes	117 (57.1)
No	88 (42.9)
Mode of oxygen administration (n = 117)	
Nasal cannula	93 (79.5)
Mask	14 (12.0)
Optiflow	10 (8.5)
Duration of oxygen (n = 117; in days; me (Q1–Q3))	1 (2–3)

**Table 2 children-11-00164-t002:** Characteristics per hospital.

Hospital	n (%)	Median Time to SCS (Q1–Q3)	Median LOS (Q1–Q3)	Median Duration of Oxygen (Q1–Q3) (n = 117)
Hospital 1	35 (17.1)	169 (75–277)	3 (2–4)	2 (1–3)
Hospital 2	38 (18.5)	144 (111.5–275.5)	3 (3–4)	2 (1–3)
Hospital 3 (2 sites)	21 (10.2)	186 (52.5–612)	3 (2–3)	2 (1–2)
Hospital 4 (2 sites)	66 (32.2)	192 (119.5–448)	4 (3–5)	2 (2–4)
Hospital 5	45 (22)	168 (69–412)	4 (3–4)	3 (2–3)

**Table 3 children-11-00164-t003:** Distribution of variables over categories of SCS administration time.

Categories of SCS Administration Time	Sex n (%)		Age m_e_ (IQR)	Reason for Admission n (%)			Medication at Home n (%)	
	Female	Male	In Months	Dyspnea	First AE	AE	Yes	No
**0** (≤60 min)	11 (39.3)	17 (60.7)	65.5 (166)	0 (0.0)	8 (28.6)	20 (71.4)	8 (28.6)	20 (71.4)
**1** (61–90 min)	9 (40.9)	13 (59.1)	87.0 (170)	2 (9.1)	5 (22.7)	15 (68.2)	5 (22.7)	17 (77.3)
**2** (91–120 min)	11 (64.7)	6 (35.3)	60.0 (160)	1 (5.9)	9 (52.9)	7 (41.2)	9 (52.9)	8 (47.1)
**3** (121–150 min)	9 (39.1)	14 (60.9)	49.0 (147)	0 (0.0)	11 (47.8)	12 (52.2)	11 (47.8)	12 (52.2)
**4** (≥151 min)	48 (41.7)	67 (58.3)	75.0 (195)	9 (5.9)	42 (36.5)	64 (55.7)	42 (36.5)	73 (63.5)

**Table 4 children-11-00164-t004:** Multivariable logistic regression model predicting the timely administration of SCS (n = 205).

Variable	β	*p*-Value	OR	95% CI (Lower–Upper)
Sex (ref. Male)	0.127	0.763	1.136	0.496–2.598
Age category (ref. age > 6 years)	0.299	0.471	1.348	0.598–3.038
Transport (ref. own)	0.661	0.442	1.936	0.360–10.414
Medication at home (ref. No)	0.425	0.343	1.530	0.635–3.686
Need for oxygen therapy (ref. No)	−0.062	0.883	0.940	0.410–2.153

**Table 5 children-11-00164-t005:** Multivariable logistic regression model predicting the delayed administration of SCS (n = 205).

Variable	β	*p*-Value	OR	95% CI (Lower–Upper)
Sex (ref. Male)	−0.127	0.763	0.881	0.385–2.015
Age category (ref. age > 6 years)	−0.299	0.471	0.742	0.329–1.672
Transport (ref. own)	−0.661	0.442	0.517	0.096–2.779
Medication at home (ref. No)	−0.425	0.343	0.653	0.271–1.574
Need for oxygen therapy (ref. No)	0.062	0.883	1.064	0.465–2.439

## Data Availability

The data presented in this study are available on request from the corresponding author. The data are not publicly available due to ethical considerations of patient data.

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
