# Peer review of "Managing Pediatric Asthma Exacerbations: The Role of Timely Systemic Corticosteroid Administration in Emergency Care Settings—A Multicentric Retrospective Study"

_children, 2024, doi:10.3390/children11020164_

Round 1
Reviewer 1 Report
Comments and Suggestions for Authors
1-Authors should consider also the risk of overtreatment of children admitted to ED for moderate exacerbation. Many children with moderate exacerbation may actually respond to higher dose of beta-2 agonist combined to inhaled corticosteroids. This is particulaly true for patients who are not receiving regular treatment of their asthma.
2-Authors should comment about the limitation of a retrospective multicenter study design. How can the Authors know if the criteria for assessing the severity of asthma exacerbation were the same between different hospitals and between different doctors in the same hospital?
3-How do the Authors explain the difference in SCS between hospital 2 (144 min) and hospital 4 (192 min) ?
Comments on the Quality of English Languagesome mispellings need to be corrected
Author Response
We thank the two reviewers and the editor for having carefully reviewed our manuscript. We believe their suggestions have substantially improved the value of our manuscript. Below, we provide a point-by-point response to their comments and indicate which changes have been made in the revised manuscript.
- Authors should consider also the risk of overtreatment of children admitted to ED for moderate exacerbation. Many children with moderate exacerbation may actually respond to higher dose of beta-2 agonist combined to inhaled corticosteroids. This is particularly true for patients who are not receiving regular treatment of their asthma.
The international guidelines clearly state that every asthma exacerbation, except for the mildest form, should be treated with systemic corticosteroids.
- Authors should comment about the limitation of a retrospective multicenter study design. How can the Authors know if the criteria for assessing the severity of asthma exacerbation were the same between different hospitals and between different doctors in the same hospital?
We thank the reviewer for this suggestion, and we accept it. Please find the addition highlighted in the manuscript.
- How do the Authors explain the difference in SCS between hospital 2 (144 min) and hospital 4 (192 min)?
This is due to the current differences in standard operating procedures in the different hospitals.
Reviewer 2 Report
Comments and Suggestions for Authors
The authors have presented a very important study, the main drawback of which is the retrospective nature.
Table 1: Please explain why such a large proportion of subjects did not receive any medication at home, even when several children had recurrent asthma admissions
Additional details are needed about the study population
1. How many children had fever at presentation?
2. How many children had wheeze or crepitations at presentation?
3. What was the initial diagnosis for all these children? Use a separate table if necessary. Please highlight how many of these children had an initial diagnosis of asthma exacerbation made within one hour of admission. Without this, SCS would not have been administered
4. how many of these children had a chest X-ray to rule out pneumonia within the first hour after admission. If diagnosis of respiratory pathology is not confirmed there is likely to be hesitation is using SCS
5. What was the algorithm of triaging and how quickly the relevant investigations were carried out in the tertiary and other secondary care centers
6. How quickly in these children the diagnosis of a recurrent admission due to asthma exacerbation was confirmed. if the previous details were not available quickly to the health workers the time taken to initiate SCS would be delayed.
All these information would be available from the medical records.
Author Response
We thank the two reviewers and the editor for having carefully reviewed our manuscript. We believe their suggestions have substantially improved the value of our manuscript. Below, we provide a point-by-point response to their comments and indicate which changes have been made in the revised manuscript.
The authors have presented a very important study, the main drawback of which is the retrospective nature.
We thank the reviewer for his comment.
- Table 1: Please explain why such a large proportion of subjects did not receive any medication at home, even when several children had recurrent asthma admissions.
We thank the reviewer for this comment. A large portion of the population were children under the age of 6 years. In this age group a firm diagnosis of asthma cannot be made, as mentioned in the discussion. This can have an impact on the physicians prescribing of medications for asthma. This can be found on page 7 line 263.
- How many children had fever at presentation?
Children with a fever were not included in the study because this could indicate that it is not really an asthma exacerbation. Only children with a confirmed asthma exacerbation were included. We added this as an exclusion criterion.
- How many children had wheeze or crepitations at presentation?
All children presented with wheezing as one of the symptoms at admission.
- What was the initial diagnosis for all these children? Use a separate table if necessary. Please highlight how many of these children had an initial diagnosis of asthma exacerbation made within one hour of admission. Without this, SCS would not have been administered.
All included children were diagnosed with an asthma exacerbation upon admission.
- how many of these children had a chest X-ray to rule out pneumonia within the first hour after admission. If diagnosis of respiratory pathology is not confirmed there is likely to be hesitation is using SCS
In Belgium, the use of chest X-rays is not standard procedure in children upon admission with respiratory symptoms that do not indicate infection.
- What was the algorithm of triaging and how quickly the relevant investigations were carried out in the tertiary and other secondary care centers?
The algorithm of triage is explained in section “design and setting”. The timing of the investigations can unfortunately not be retrieved from the EMR.